

# Metabolite profiling of endophytic *Streptomyces* spp. and its antiplasmodial potential

Siti Junaidah Ahmad[1,2], Noraziah Mohamad Zin[2], Noor Wini Mazlan[3], Syarul Nataqain Baharum[4], Mohd Shukri Baba[5] and Yee Ling Lau[6]

[1] Faculty of Health Sciences, University of Sultan Zainal Abidin, Kuala Nerus, Terengganu, Malaysia
[2] Center for Diagnostic, Therapeutic and Investigative Studies, Faculty of Health Sciences, Universiti Kebangsaan Malaysia, Kuala Lumpur, Malaysia
[3] Analytical and Environmental Chemistry, Faculty of Science and Marine Environment, Universiti Malaysia Terengganu, Kuala Nerus, Terengganu, Malaysia
[4] Institute of Systems Biology, Universiti Kebangsaan Malaysia, Bangi, Selangor, Malaysia
[5] Department of Biomedical Science, Kulliyyah of Allied Health Sciences, International Islamic University, Kuantan, Pahang, Malaysia
[6] Department of Parasitology, Faculty of Medicine, Universiti Malaya, Kuala Lumpur, Malaysia

Corresponding authors
Siti Junaidah Ahmad,
junaidahahmad@unisza.edu.my
Noraziah Mohamad Zin,
noraziah.zin@ukm.edu.my

## ABSTRACT

**Background:** Antiplasmodial drug discovery is significant especially from natural sources such as plant bacteria. This research aimed to determine antiplasmodial metabolites of *Streptomyces* spp. against *Plasmodium falciparum* 3D7 by using a metabolomics approach.
**Methods:** *Streptomyces* strains' growth curves, namely SUK 12 and SUK 48, were measured and *P. falciparum* 3D7 $IC_{50}$ values were calculated. Metabolomics analysis was conducted on both strains' mid-exponential and stationary phase extracts.
**Results:** The most successful antiplasmodial activity of SUK 12 and SUK 48 extracts shown to be at the stationary phase with $IC_{50}$ values of 0.8168 ng/mL and 0.1963 ng/mL, respectively. In contrast, the $IC_{50}$ value of chloroquine diphosphate (CQ) for antiplasmodial activity was 0.2812 ng/mL. The univariate analysis revealed that 854 metabolites and 14, 44 and three metabolites showed significant differences in terms of strain, fermentation phase, and their interactions. Orthogonal partial least square-discriminant analysis and S-loading plot putatively identified pavettine, aurantioclavine, and 4-butyldiphenylmethane as significant outliers from the stationary phase of SUK 48. For potential isolation, metabolomics approach may be used as a preliminary approach to rapidly track and identify the presence of antimalarial metabolites before any isolation and purification can be done.

## INTRODUCTION

A large number of imported malaria cases from Europe and the Mediterranean have recently been reported. The increasing number of foreign travelers linked to the hefty inflow of malaria-endemic immigrants. For instance, anopheline vectors, which act as the

parasite reservoir and are present in Mediterranean regions, can infect the travelers returning from tropical countries (*Dominguez Garcia et al., 2019*; *Odolini, Gautret & Parola, 2012*). In 2017, World Health Organization (WHO) documented that the most malaria cases during that year were reported in Africa (with 92% or 200 million), and followed by Southeast Asia (with 5% or 10.8 million) (*WHO, 2018*).

Endophytic *Streptomycetes* are bacteria within ethnomedicinal plants that have symbiotic relationships with the host plants. *Streptomyces* is the biggest genus of actinomycetes, consisting of aerobic Gram-positive bacteria that are capable of generating filamentous branches called mycelia (*Shirling & Gottlieb, 1966*). The mycelia secrete antibiotics, that may be anti-fungal, antibacterial, and anti-viral, when they sporulate in the dormant phase (*Alam et al., 2010*; *Gramajo, Takano & Bibb, 1993*; *Roszak & Colwell, 1987*).

Previous studies reported that trioxacarcin A and D, produced by marine *Streptomyces* sp., were more effective in producing antiplasmodial activity than artemisinin (*Maskey et al., 2004*; *Tomita et al., 1981*). Furthermore, coronamycin, a novel antiplasmodial agent from *Streptomyces* sp., reportedly possessed antiplasmodial activity against *Plasmodium falciparum* (*Ezra et al., 2004*). Strobel and co-researchers discovered that munumbicin D also had antiplasmodial activity (*Castillo et al., 2002*). Kakadumycin A, also identified by the same team, isolated from *Streptomyces* sp. NRRL 30566 compound demonstrated promising antiplasmodial activity on *Plasmodium falciparum* (*Castillo et al., 2003*). Furthermore, the isolated gancidin W of *Streptomyces* sp. SUK 10 inhibited the growth of *Plasmodium berghei* (*Zin et al., 2017*).

This study aims to determine the antiplasmodial activity on metabolites produced from *Streptomyces* spp. against *Plasmodium falciparum* 3D7 using a metabolomics approach. Metabolomics is a high throughput method utilized to screen the metabolites in organisms or tissues by chromatography techniques coupled with mass spectrometry (MS) (*Rochfort, 2005*). Then, the metabolites are depicted into a two-dimensional distribution using multivariate analysis, including Principal Component Analysis (PCA) and Orthogonal Projections to Latent Structures Discriminant Analysis (OPLS-DA) and further putatively identified using Dictionary of Natural Products (DNP). The metabolites include amino acids, carbohydrates, organic acids, vitamins, antibiotics and phytochemicals (*Wishart, 2008*). The metabolomics approach uses high analytical techniques to determine metabolites in the biological samples. In this study, a hybrid chemical profile using liquid chromatography and mass spectrometry (LC–MS) with multivariate data analysis was used to determine the metabolites present in *Streptomyces kebangsaanensis* SUK 12 and SUK 48. Moreover, MS-based metabolomics helped to fast-track the identification of targeted and untargeted metabolites present in complex extracts during screening.

## MATERIALS AND METHODS

### Culture

*Plasmodium falciparum* 3D7 was obtained from the culture collection of the Parasitological Department, Faculty of Medicine, Universiti Malaya while *Streptomyces kebangsaanensis* SUK 12 and SUK 48 were acquired from the Novel Antibiotics Laboratory, Programme of Biomedical Science, Faculty of Health Sciences, Universiti

Kebangsaan Malaysia. *Streptomyces kebangsaanensis* SUK 12 was isolated from an ethnomedicinal plant, *Portulaca oleracea* L., which was collected from the Nenasi Reserve Forest, Pahang, Malaysia (*Sarmin et al., 2013*). *Streptomyces* sp. SUK 48 was isolated from the fruit of *Brasilia* sp. (*Zin et al., 2015*).

### *Streptomyces* spp. growth condition

The SUK 12 and SUK 48 strains were grown in nutrient broth (pH 7.0) on an orbital shaker at 160 rpm, 28 °C for 21 days. The dry weight was collected daily by centrifugation at 4,000 rpm for 15 mins at 25 °C and dried at 70 °C. The growth curves of both strains were plotted using Microsoft Excel. The growth rate and generation time were estimated by the following calculation:

$$\text{Growth rate, } k = \frac{\log(\text{Higher dry weight, } X1) - \log(\text{lower dry weight, } X0)}{0.301t \text{ (time between two intervals)}}$$

$$\text{Generation time} = \frac{1}{k}$$

### *Streptomyces* spp. extracts preparation

The crude extracts prepared in the 14-day culture of both strains' blocks (1 cm$^2$) were used to inoculate in nutrient broth for 5, 12 and 14 days for SUK 12, and 7, 14 and 21 days for SUK 48. The broth cultures were also incubated on an orbital shaker (Protech, Malaysia) at 160 rpm, 28 °C. Both strains' cultures were extracted using 3-fold ½ volume of ethyl acetate. The organic layer (ethyl acetate) was discarded and dried under vacuum using a rotary evaporator (Buchii, Swirtzerland).

### In vitro antimalarial assay

The 96-well plate in vitro antimalarial activity includes a series of extracts dilution (complete media or complete medium (CM) and extracts), positive drug control (CM and CQ), positive control (CM plus iRBC, infected red blood cells (RBC)), and negative control (CM plus fresh RBC). The positive control was the maximum parasite lactate dehydrogenase (pLDH) enzyme absorbed in cells, and the negative control was a blank (*Lambros & Vanderberg, 1979*).

### *Plasmodium falciparum* 3D7 culture

The volume of iRBC was calculated then a complete media (containing RPMI 1,640, 2.3 g/L sodium bicarbonate, 4 g/L dextrose, 5.957 g/L HEPES, 0.05 g/L hypoxanthine, 5 g/L Albumax II, 0.025 g/L gentamycin sulfate, 0.292 g/L L-glutamine) and fresh RBC were added into the culture to make the final values of 1% parasitaemia and 2% hematocrit. The parasitaemia level was measured using a thin blood film (TBS) that was stained with 10% Giemsa and observed under a light microscope with 1,000×$g$ magnification. Next, when the parasitaemia level reached within 5–7%, iRBC was synchronized with 5% sorbitol to obtain the ring stage of the parasite (*Amir et al., 2016*; *Lambros & Vanderberg, 1979*).

## Dilution of *Streptomyces* spp. extracts

*Streptomyces* spp. extracts stock solution was prepared using 10 mg/mL dimethyl sulfoxide (DMSO) and CM. The final concentration of the prepared DMSO was less than 1% to prevent its toxic effect on the parasite. Next, the stock solution of *Streptomyces* spp. extracts was serially tenfold-diluted eight times (starting at 1,000 µg/mL and ending at 0.0001 µg/mL). For the control, the drug chloroquine diphosphate (CQ) was used in various dilutions ranging from 10 µg/mL to $10^{-6}$ µg/mL. Next, 100 µL of the diluted extracts and CQ were transferred into a sterile 96-wells plate. For negative and positive control wells, 100 µL of CM was transferred into this 96-well plate.

## Incubation of *Streptomyces* spp. extracts with parasite

Parasite culture (iRBC) with 10% parasitaemia was selected and centrifuged at 1,800 rpm for 5 mins to obtain the cell pellets. The cell pellets were diluted to 2% parasitaemia with fresh RBC. Approximately 2 µL iRBC was transferred into every well of diluted extracts, control drug, CQ and positive control. Then, 2 µL of fresh RBC was added into negative control, and the 96-well was incubated at 37 °C, 5% carbon dioxide ($CO_2$) for 48 h. The plate was then frozen overnight at −20 °C prior the pLDH assay started (*Makler et al., 1993*).

## pLDH assay

Upon overnight freezing at −20 °C, the 96-well plate was defrosted at 37 °C for 20 min and re-frozen at −20 °C for 30 min. This step was repeated 3 times to break the parasite cells. At the same time, Malstat reagent and NBT-PES (nitroblue tetrazolium-phenazine ethosulfate) were also prepared in the dark. About 100 µL of Malstat reagent and 25 µL of NBT-PES were added to a new 96-well plate. After the freezing and defrosting processes finished, 15 µL of each defrost culture well plate was transferred into the wells of a new plate that contained Malstat reagent and NBT-PES. The 96-well plate was then incubated for 1 h at room temperature in the dark. Absorbance readings at the wavelength of 650 nm ($A_{650}$) were measured for the 96-well plate using a spectrophotometer (M200; Tecan, Männedorf, Switzerland). The positive control was assumed to be the maximum level of lactate dehydrogenase enzyme production. The negative control was the blank (*Makler et al., 1993*; *Trager & Jensen, 1976*). Inhibition of parasite (%) was calculated as follows:

$$\frac{(A650 \text{ average for diluted sample} - A650 \text{ average negative control}) \times 100}{(A650 \text{ average positive control} - A650 \text{ average negative control})}$$

A sigmoid graph was plotted using Graphpad PRISM version 7. The value of parasite inhibition 50% ($IC_{50}$) was determined from the graph.

## LC–MS analysis

Approximately 3 mg/mL extracts of S12D5 (SUK 12 day to harvest fermented broth is 5), S12D12, S48D7 and S48D14 were sent for LC–MS analysis (*Rosli et al., 2017*). Scientific C-18 column Thermo was chromatographically separated by an UltiMate 3000 UHPLC (Dionex, Sunnyvale, CA, USA) system (AcclaimTM Polar Advantage II, 3–150 mm,

3 µm particle size). A flow rate of 0.4 mL/min at 40 °C was maintained with water that contained 0.1% formic acid and 100% acetonitrile with a total operating time of 22 min. The gradient started for 3 min at 5% of solvent B, then grew to 80% of solvent B for 7 min and stayed at 80% of solvent B for 5, or 10–15 min. At last, the gradient turned to 5% of solvent B in 7 min (15–22 min). The ESI-positive ionization was performed with the use of MicrOTOF-Q III (Bruker Deltonic, Billerica, MA, USA) with the settings: capillary voltage 4,500 V; pressure of 1.2 bar, and dry gas flow 8 L/min at 200 °C; 50–1,000 Da m/z, respectively.

## MS data handling

Raw material in "d format" was supplied to Bruker Compass Data Analysis Viewer version 4.2 (Bruker Daltonics, Bremen, Germany) and imported into the Profile Analysis 2.0 data bucketing software (Bruker Daltonics, Bremen, Germany) (*Mamat et al., 2018*; *Veeramohan et al., 2018*). The parameters for compound detecting were: signal/noise threshold: 5; correlation coefficient: 0.7; minimum compound length: 8; smoothing width: 2. Compound detection was done using Find Molecular Features (FMF). The composite bucket table was calculated using advanced bucketing features as time alignment parameters. The time interval was between 0.00 min and 22.04 min, and the mass range was between 49 m/z and 1,001 m/z. For standardized settings, the data was uploaded to the MetaboAnalyst 3.0 server (www.metaboanalyst.ca). The normalizing features used were: internal standard caffeine: 195.088 m/z, RT: 7.98 min; transformation: log; scaling: pareto. (*Xia et al., 2015*). Normalized and validated data tables were exported to SIMCA P+ version 15 from Umetrics AB, Umea, Sweden. The PCA, the examination of fundamental variations in samples, and the presentation of outliers were conducted prior to the sample classification of Model OPLS-DA. With 100 random permutations using cross-validations and responsive permutation tests, the robustness of OPLS-DA was validated. The two-way ANOVA type was "ANOVA" in subjects. The significance threshold was set as the $p$-value of correction lower than 0.05. The false discovery rate was less than 0.05.

## Metabolite identification

The search and manual verification of secondary metabolites in the LC–MS analyses were done. Online databases, namely METLIN and DNP, were examined for the value of a molecular ion of interest (*Mazlan et al., 2019*; *Smith et al., 2005*). The databases were used to identify molecular weight metabolites within a specified m/z value query tolerance range.

## DNA extraction

First, neutral lysis of genome DNA was done for isolation, followed by extraction, then precipitation of phenol chloroforms and isopropanol. The protocol was used to extract the DNA (*Kieser, 1984*). For several changes, *Streptomyces* mycelium, in a 1.5 mL of Eppendorf tube was interrupted by vortexing with 500 µL of lysozyme. The samples were then added with 25 µL of 50 mg/mL of lysozyme and 3 µL of 10 mg/mL of RNase. Finally, the

purity and concentration of the DNA were determined on a Thermo Scientific nano-drop 2000C machine. SUK 12 DNA sequence was obtained from the previous study (*Sarmin et al., 2013*).

### 16S rRNA molecular identification and phylogenetic tree analysis

Polymerase chain reaction amplified the 16S rRNA gene sequence using universal 16SrRNA bacterial gene primers as described earlier (*Coombs & Franco, 2003*). For SUK 12 and SUK 48, 1,416 and 1,463 nucleotides respectively, were near full-length 16S rRNA gene sequences. The sequences were compared using BLAST and EzTaxon e-databases with the GenBank database (*Altschul et al., 1997*). The comprehensive sequence similarity calculation was based on the EzTaxon server's global alignment algorithm (*Kim et al., 2012*). The sequence was also matched several times with 16S rRNA gene sequences, which are available in GenBank/EMBL with the CLUSTAL W programme, for closely related species of *Streptomyces* (*Thompson, Higgins & Gibson, 1994*). The neighbour-joining (NJ) approach (*Saitou & Nei, 1987*), available in MEGA software package 7 version, used the reconstructions of the phylogenetic trees (*Tamura et al., 2013*). In accordance with the two-parameter Kimura model, the matrices of distance were calculated (*Kimura, 1980*). Moreover, the tree was made with the maximum likelihood (ML) (*Kimura, 1980*). By conducting a bootstrap analysis-based of 1,000 replicates, the topology of the trees was evaluated (*Felsenstein, 1985*).

## RESULTS

### Growth curve and effectiveness of antiplasmodial activity of *Streptomyces* spp

The growth curve showed that the generation time of *Streptomyces* sp. SUK 12 was faster than SUK 48's with 1.16 h/generation and 4.56 h/generation, respectively (Figs. 1A and 1B). Meanwhile, the antiplasmodial activity revealed that the crude extract of *Streptomyces* sp. SUK 48 on day 14 was more potent (with an $IC_{50}$ value of 0.1963 ng/mL) than that of SUK 48 on day 7 and SUK 12 on day 5 and day 10 (Table 1).

### Metabolomics approach

The metabolite profiles in extracts using two-way ANOVA (univariate analysis) were compared to those using PCA, OPLS-DA and S-Plot (multivariate analysis). Two-way ANOVA summarizes and simplifies each sample metabolites (Fig. 2). Approximately, 96 metabolites were common in the type of strain, fermentation time, and interaction between both (time and strain type). While 44 metabolites were significant in strain type, 14 metabolites were significant in fermentation time, and three metabolites were significant in the interaction between both (Table S1). The variation and diversity of the extracts were examined in a PCA model (Fig. 3A). The PCA model, however, gave a low $Q2 = 0.271$ preview. *Streptomyces* sp. extracts cannot be distinguished by any significant variation. The OPLS-DA scores scatter plot of the crude extracts SUK 48 showed distinct separation between day 7 and day 14 (Fig. 3B). Nevertheless, the crude

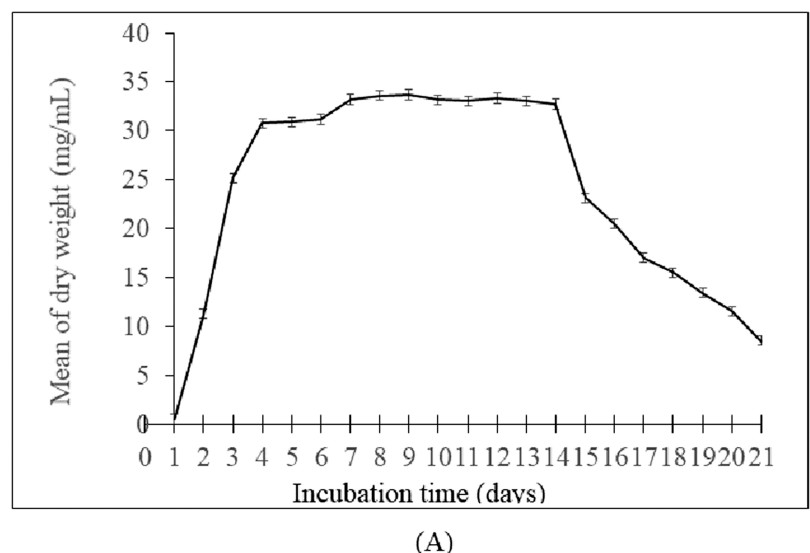

(A)

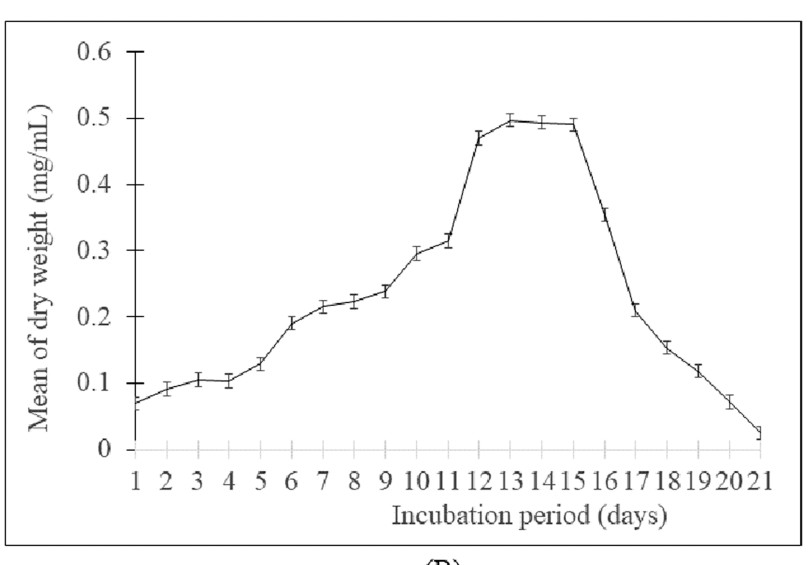

(B)

**Figure 1 Growth curve of *Streptomyces* spp.** (A) The growth curve of *Streptomyces* sp. SUK 12 and (B) SUK 48.

**Table 1 Anti-plasmodial activity of *Streptomyces* spp. extracts.**

| Sample (growth phase) | Inhibitory concentration ( IC$_{50}$ in ng/mL ± SEM) |
|---|---|
| S12D5 (mid exponential phase) | 18.62 ± 0.00 |
| S12D12 (stationary phase) | 0.8168 ± 0.174 |
| S12D14 (death phase) | 62.29 ± 0.00 |
| S48D7 (mid exponential phase) | 0.1980 ± 0.099 |
| S48D14 (stationary phase) | 0.1963 ± 0.17 |
| S48D21 (death phase) | 527.4 ± 0.00 |
| Chloroquine diphosphate (control drug) | 0.2821 ± 0.00 |

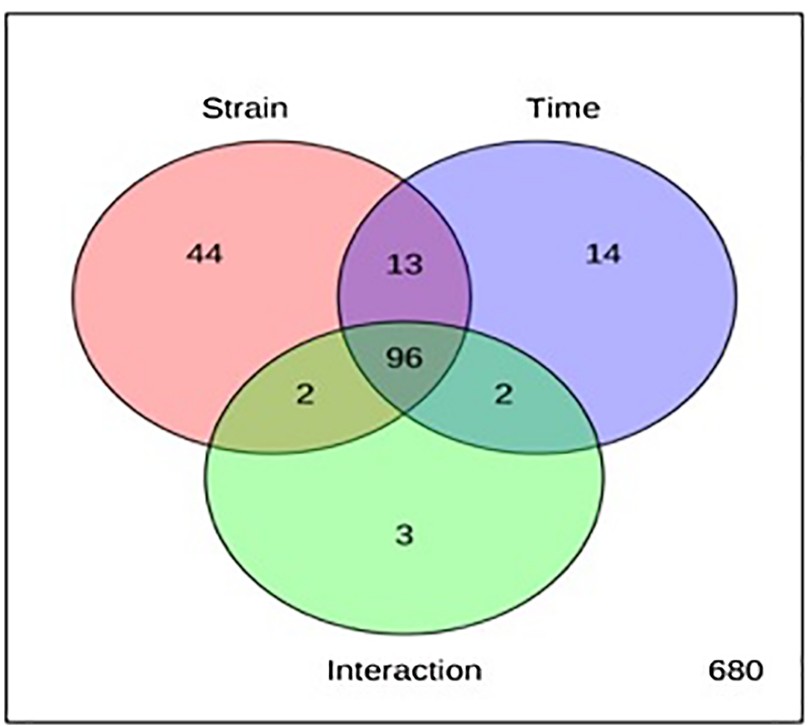

**Figure 2 Two-way analysis of variance (ANOVA).** The red circle with 155 metabolites represents type of strain, SUK 12 and SUK 48, whereas blue circle with 125 metabolites represents fermentation time and green circle with 103 metabolites represents interaction between both (time and strain type).

extract of SUK 12 on day 5 and day 12 was not well distributed. Meanwhile, the OPLS-DA scores scatter plot (Fig. 3C) and S-loadings plot (Fig. 3D) of SUK 48 crude extracts between day 7 and day 14 gave a good distribution. Overall, the metabolites separation from SUK 48 day 14 extracts could be seen as in Fig. 4. The secondary metabolite at m/z 195.092 was putatively identified as 1-Vinyl-β-carboline (pavettine) whereas metabolites at m/z 225.163 and 225.153 were putatively identified as 4-butyldiphenylmethane. Metabolites presence at m/z 227.135, 227.145, and 227.153 were putatively identified as aurantioclavine; (-)-form. All the outliers in the crude extracts of SUK 48 day 14 mentioned as above are known metabolites in Table 2. Then, metabolites at m/z 211.135 and 211.123 were putatively identified as 2,5-Dimethyl-3-(2-phenylethenyl)pyrazine, while metabolites at m/z 245.117 and 245.137 were 3,3-Bis(4-hydroxyphenyl)-1-propanol. The secondary metabolite at m/z 195.149 was represented as 6-(1-Methylethyl)-3-(2-methylpropyl)-2(1H)-pyrazinone, while at m/z 155.078 was trifluoromethyl piperazine. These metabolites were the outliers in SUK 48 day 7 (Table 3).

### *Streptomyces* spp. identification using molecular analysis

Phylogenetic tree analysis of both SUKs showed that SUK 48 and SUK 12, using the NJ tree, belonged to different clades (Fig. 5) where the bootstrap value was 98%. Moreover, the ML tree (Fig. S1) supported this report with a bootstrap value of 95%.

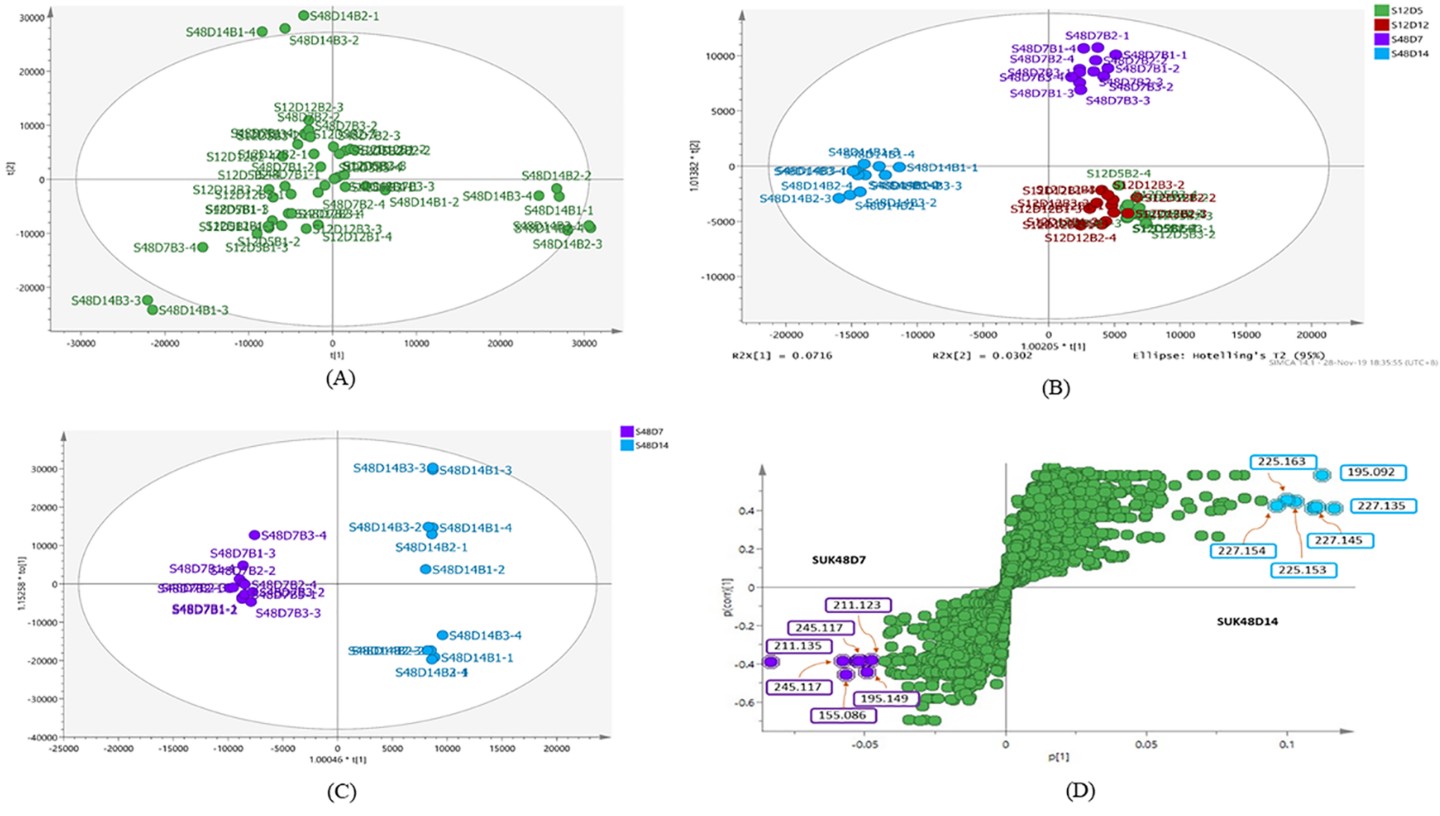

**Figure 3 Principal component analysis.** (A) PCA scores scatter plot of SUK 12 (day 5 and 12) and SUK 48 (day 7 and 14) crude extracts ($R2 = 0.47$; and $Q2 = 0.271$). (B) OPLS-DA scores scatter plot between SUK 12 and SUK 48 crude extracts ($R2(Y) = 0.651$; $Q2 = 0.395$). (C) OPLS-DA scores scatter plot between S48D7 and S48D14, ($R2(Y) = 0.996$; $Q2 = 0.61$). (D) S-plot of SUK 48 day 7 vs. day 14 metabolites.

## DISCUSSION

There are four stages of bacteria growth: lag, log, stationary, and death. The crucial phase for producing antibiotics is the stationary phase in *Streptomyces* spp. (*Chandrakar & Gupta, 2019*; *Chen et al., 2020*). *Streptomyces lactamdurans* that produces the antibiotic cephamycin C has a generation time of 7.5 h (*Ginther, 1979*). Furthermore, faster growth is potentially essential in the enzyme production of actinomycetes. The effective mycelial fragmentation by the enhanced expression SsgA has significant consequences for antibiotic production, with increasing undecylprodigiosin, but a complete block in the production of actinorhodin (*Van Wezel, McKenzie & Nodwell, 2009*). SUK 48 could therefore be a potential antibiotic producer as its ability as slow-growing capacity. The stationary phase of bacterial growth is the survival phase in which they actively secrete secondary metabolites to combat oxidative stress and malnutrition for survival purposes (*Banchio & Gramajo, 2002*; *Undabarrena et al., 2017*).

Chemometrics is a chemical discipline that uses mathematical and statistical logic-based methods to design optimal measurement procedures and tests, and provide maximum chemical information through the analysis of chemical data (*Massart & Buydens, 1988*). From a metabolomics perspective, the univariate analysis is a statistical test used to

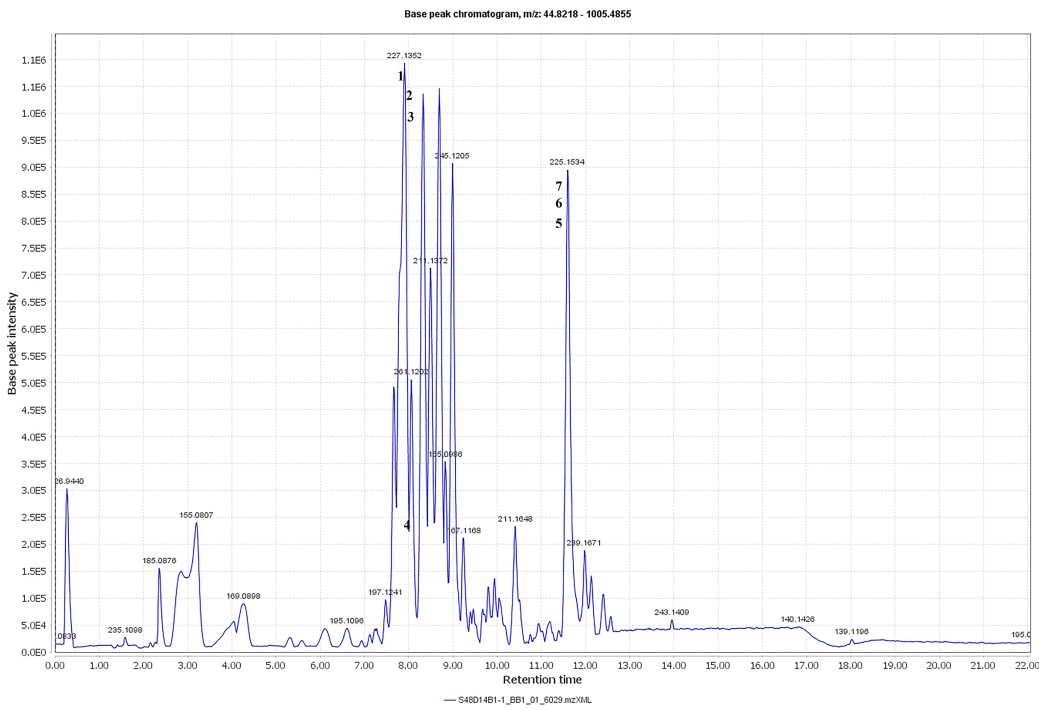

**Figure 4 Total ion chromatogram.** Total ion chromatogram of the crude extract of SUK48 on day 14 with labelled of significant peaks of outliers (1–6).

independently measure metabolites that have significantly increased or decreased between different groups. This test is important for significance testing of the tens to hundreds of metabolites to reduce the probability of false positives caused by multiple tests. Of the 854 metabolites analyzed in two-way ANOVA, 14 were significant in the fermentation time (mid-exponential and stationary phase), whereas, 44 and 14 metabolites were significant in strain and time respectively, and interaction between the both were three metabolites were significant. In addition, there were 96 metabolites significant in strain, time, and interaction of both. There were two antimalarials identified in the list of 96 metabolites, gancidin W, antimycin and hydroquinine. Gancidin W has been reported isolated from *Streptomyces gancidicus* and poses antimalarial agent against *Plasmodium berghei* (*Aiso, Suzuki & Takamizawa, 1956*; *Wakaki et al., 1958*; *Zin et al., 2017*). Besides, antimycin had been reported was isolated from *Streptomyces blastmyceticus* and antiplasmodial activity against *Plasmodium falciparum* K1 (*Gomez-Lorenzo et al., 2018*; *Nakayama, Okamoto & Harada, 1956*). While hydroquinine was first isolated from the reduction of quinine and was determined as an antimalarial agent (*Buttle et al., 1938*; *Griffin et al., 2012*). Pavettine is an alkaloid compound which found in SUK 48 on day 14, was reported to have anti-cancer where $IC_{50}$ was 100 ng/mL against leukemic cell lines (*Figuerola & Avila, 2019*). Furthermore, the antimicrobial activity of pavettine against *Bacillus subtilis* and *Candida albicans* reported minimum inhibitory concentration (MIC) values from 1.9 μg/disc to 3.8 μg/disc (*Sudha & Masilamani, 2012*). Pavettine was

**Table 2 Identification of significant metabolites of SUK48 day 14 extracts.** Putatively identified metabolites using DNP and METLIN that presence in the crude extracts of SUK48 day 14 highlighted S-Plot.

| No | m/z value | Retention time (minutes) | Molecular weight | Monoisotopic mass | Compound name | Tolerance (ppm) | Chemical formula | Sources |
|---|---|---|---|---|---|---|---|---|
| 1 | 227.154 | 7.91 | 226.1457 | 226.1470 | Aurantioclavine; (−)-form | −0.3401 | $C_{15}H_{18}N_2$ | *Penicillium aurantiovirens* |
| 2 | 227.145 | 7.91 | 226.1377 | 226.1470 | Aurantioclavine; (−)-form | −0.3401 | $C_{15}H_{18}N_2$ | *Penicillium aurantiovirens* |
| 3 | 227.135 | 7.91 | 226.1277 | 226.1470 | Aurantioclavine; (−)-form | −0.3401 | $C_{15}H_{18}N_2$ | *Penicillium aurantiovirens* |
| 4 | 195.092 | 7.94 | 194.0844 | 194.0844 | 1-Vinyl-β-carboline (pavettine) | 0.02 | $C_{13}H_{10}N_2$ | *Pavetta lanceolata, Cribricellina cribraria, Costaticella hastata* and *Soulamea raxinifolia* |
| 5 | 225.163 | 11.60 | 224.1565 | 224.1565 | 4-Butyldiphenylmethane | 3 | $C_{17}H_{20}$ | Chemically synthesized |
| 6 | 225.153 | 11.60 | 224.1460 | 224.1460 | 4-Butyldiphenylmethane | 3 | $C_{17}H_{20}$ | Chemically synthesized |

**Table 3 Identification of significant outliers from SUK48 day 7 extracts.** Putatively identified metabolites using DNP and METLIN that presence in the crude extracts of SUK48 day 7 highlighted S-Plot.

| No | m/z value | Retention time | Molecular Weight | Monoisotopic mass | Compound Name | Tolerance (ppm) | Chemical Formula | Sources |
|---|---|---|---|---|---|---|---|---|
| 1. | 211.135 | 8.68 | 210.1353 | 210.1157 | 2,5-Dimethyl-3-(2-phenylethenyl)pyrazine | −1.7585 | $C_{14}H_{14}N_2$ | *Iridomyrmex humilis* |
| 2. | 211.123 | 8.68 | 210.1153 | 210.1157 | 2,5-Dimethyl-3-(2-phenylethenyl)pyrazine | −1.7585 | $C_{14}H_{14}N_2$ | *Iridomyrmex humilis* |
| 3. | 245.117 | 8.98 | 244.1099 | 244.1099 | 3,3-Bis(4-hydroxyphenyl)-1-propanol | −0.1998 | $C_{15}H_{16}O_3$ | *Streptomyces albospinus* 15-4-2 |
| 4. | 245.137 | 8.98 | 244.1298 | 244.1099 | 3,3-Bis(4-hydroxyphenyl)-1-propanol | −0.1998 | $C_{15}H_{16}O_3$ | *Streptomyces albospinus* 15-4-2 |
| 5. | 195.149 | 10.47 | 194.1415 | 194.1419 | 6-(1-Methylethyl)-3-(2-methylpropyl)-2(1H)-pyrazinone | −2.1236 | $C_{11}H_{18}N_2O$ | *Aspergillus flavus* |
| 6. | 155.078 | 3.20 | 154.0718 | 154.0718 | 2-(Trifluoromethyl)piperazine | 6 | $C_5H_9F_3N_2$ | Chemically synthesized |

previously mainly found in *Pavetta lanceolata, Cribricellina cribraria, Costaticella hastata,* and *Soulamea raxinifolia* (*Blackman & Walls, 1995*; *Jordaan, Du Plessis & Joynt, 1968*). Further isolation and purification of putative metabolites and identification and elucidation of molecule structure is required by using MS/MS, 1D- and 2D-NMR.

Phylogenetic tree analysis of both SUKs showed that SUK 48 and SUK 12 belonged to different clades. The reliability test was over 70%, which was 98% at nodes between SUK 12 and SUK 48. This result suggested that both SUKs were different subspecies. Distance matrices of the neighbouring tree construction were calculated using Kimura, the two-parameter correction model (*Kimura, 1980*), and a bootstrap analysis based on 1,000 replications was performed to evaluate the topology of the neighbouring joining trees (*Felsenstein, 1985*). SUK 12 was a novel species (*Sarmin et al., 2013*) with the antiplasmodial agent (*Remali, 2016*). We believe SUK 48 is presumably a novel species (data not published). The selected outgroup, *Kitasatospora setae* KM-6054[T] was used based on the previous study (*Sarmin et al., 2013*).

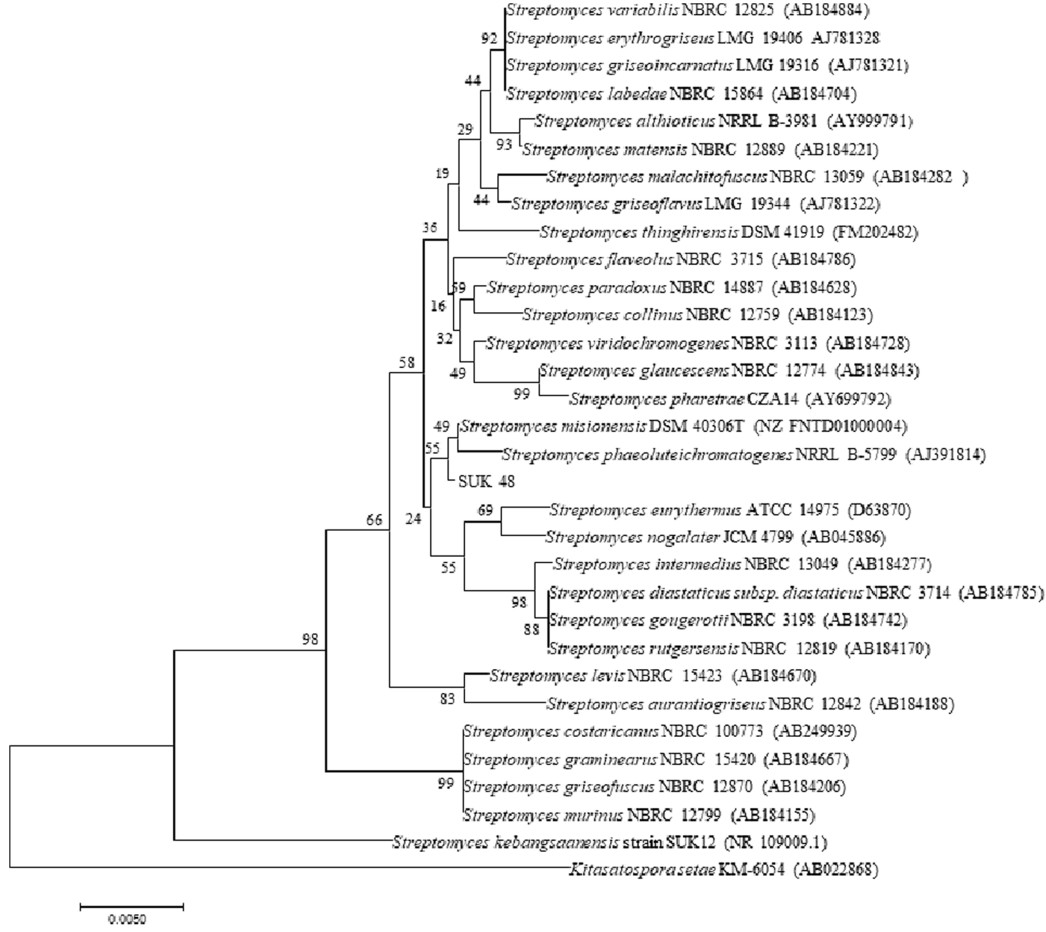

**Figure 5 Phylogenetic tree.** Phylogenetic tree of full length 16S rRNA nucleotide sequences using Neighbour Joining method showing the relationship of strain SUK12 and SUK 48 with closely related members of the genus *Streptomyces* and *Kitasatospora setae* KM-6054[T] as the outgroup. Numbers at the nodes indicate levels of bootstrap support based on 1,000 replication. Bar, 0.005 changes per nucleotide.

## CONCLUSION

SUK 48 at the stationary phase produced more metabolites with antimicrobial and anticancer properties than SUK 48 at the mid-exponential phase. Therefore, SUK 48 at the stationary phase was the potential antimalarial extract candidate for the further purification process. The metabolomics approach is a rapid tool that dereplicated known antimalarial metabolites and determined those at specific fermentation time. Further study will focus on the fractionation and purification of possible antimalarial compounds from *Streptomyces* spp. SUK 48. Meanwhile, an in silico study using molecular docking (MD) will be conducted to verify the potential interaction between complex protein-ligands in developing a potential new antiplasmodial agent.

## ACKNOWLEDGEMENTS

The authors would like to thank Jonathan Wee Kent Liew for supplying the *Plasmodium falciparum* 3D7 culture and providing coaching and guidance, and Dr. RuAngelie

Edrada-Ebel for granting permission to access the Dictionary of Natural Products (DNP) database library.

### Funding
This work received financial support from the Ministry of Higher Education via FRGS/1/2016/STG05/UKM/02/5 grant. We also received the Centre for Research and Instrumentation Management (CRIM), UKM for Research Instrumentation Development Grants awarded in 2010 and 2013 (PIP-CRIM) and Dana Modal Insan (MI-2018-004). The funders had no role in study design, data collection and analysis, decision to publish, or preparation of the manuscript.

### Grant Disclosures
The following grant information was disclosed by the authors:
Ministry of Higher Education via: FRGS/1/2016/STG05/UKM/02/5.
Centre for Research and Instrumentation Management (CRIM).
Research Instrumentation Development Grants awarded in 2010 and 2013 (PIP-CRIM).
Dana Modal Insan: MI-2018-004.

### Competing Interests
The authors declare that they have no competing interests.

### Author Contributions
- Siti Junaidah Ahmad conceived and designed the experiments, performed the experiments, analyzed the data, prepared figures and/or tables, and approved the final draft.
- Noraziah Mohamad Zin conceived and designed the experiments, authored or reviewed drafts of the paper, and approved the final draft.
- Noor Wini Mazlan analyzed the data, prepared figures and/or tables, authored or reviewed drafts of the paper, and approved the final draft.
- Syarul Nataqain Baharum conceived and designed the experiments, authored or reviewed drafts of the paper, and approved the final draft.
- Mohd Shukri Baba conceived and designed the experiments, authored or reviewed drafts of the paper, and approved the final draft.
- Yee Ling Lau conceived and designed the experiments, authored or reviewed drafts of the paper, and approved the final draft.

### Data Availability
Raw data metabolomics are available in a Supplemental File.

### Supplemental Information
Supplemental information for this article can be found online at http://dx.doi.org/10.7717/peerj.10816#supplemental-information.

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
