# Peer review of "Metabolite profiling of endophytic Streptomyces spp. and its antiplasmodial potential"

_PeerJ, doi:10.7717/peerj.10816_

## Round 0.1 · original submission · Major Revisions

Dear Dr. Ahmad and colleagues:

Thanks for submitting your manuscript to PeerJ. I have now received two independent reviews of your work, and as you will see, one reviewer raised serious concerns about the research and recommended rejection. The other reviewer was only a bit more enthusiastic, raising some serious concerns. I agree with the concerns of both reviewers.

Fortunately, there is a lot of criticism here for you to consider. If you choose to address these concerns, I encourage you to revise your work and resubmit to PeerJ (I will be happy to handle your manuscript again). However, I strongly encourage you to take into account all of the concerns raised by both reviewers.

Thanks again for submitting your work to PeerJ.

Good luck with your research moving forward,

-joe

Reviewer 1 ·

Basic reporting

Generally, this paper is adequately sourced with appropriate references.
1. English language usage needs careful revision throughout
2. Legibility of figures must be improved. Figures, especially 4, 5, and 6, are blurry with small text that makes them illegible.
3. Supplementary table 1: as the compounds listed are only putative matches for ions and not confirmed, the M/Z values and retention times should be provided to allow these compounds to be identified. The type of analysis yielding this list should also be mentioned in the table caption.
4. supplemental metabolomics raw data Excel file: data need labels. Presumably these are ion abundances?
5. tables 1 and 2: unit needed for retention time, exact monoisotopic mass should be provided instead of molecular weight

Experimental design

This paper is a step toward the discovery of novel antimalarial compounds and excitingly reports extracts of bacterial cultures with antimalarial activity. Producing strains are identified as Streptomyces by DNA sequencing. Mass spectrometry of extracts is used to identify putative antimalarial compounds and metabolomics analysis is used, though the goal of this analysis is unclear.
1. The goal of the metabolomics analysis (ANOVA, PCA, OPLS-DA) is not clear. The authors should clearly explain what they mean by "prioritizing [the] most effective extract" (line 227). Are they prioritizing individual extracts based on uniqueness? Are they trying to correlate individual metabolites with observed activity? Presumably the ultimate goal will be identifying the individual compounds responsible for antimalarial activity in these extracts, but it's not clear how this metabolomics analysis advances that goal. Also, if the goal of metabolomics is dereplication rather than prioritization, this should be reflected in the title of the paper.
2. It's not clear why metabolites from day 14 and day 7 cultures of SUK 48 were chosen for dereplication (lines 241-255). A justification for prioritizing these extracts should be given. It's also not clear why these particular metabolites from these samples were chosen, nor how these ions were identified as [M + H]+ ions. These issues should be rectified.
3. Justification for analyzing these particular samples needs to be presented. Some form of sample collection or strain information should be supplied. If these are of interest because they are endophytic, what is the host plant?
4. The data in Figure 2 should be presented in a table rather than a bar graph. Also, A justification for evaluating each strain using a different series of time points should be provided. Unless there is a compelling reason, both should be evaluated using the same time course.
5. Line 222: method for quantifying generation times needs to be stated
6. Mass spectrometry data handling methods reference a "missing value" (line 176), but we don't know which group that was for which value is missing.
7. Rephrase to clarify inoculation and culturing protocol lines 93-94: unclear whether there were multiple inoculations at different time points.
8. figure 6: how were these particular strains chosen for comparison? Was there a similarity cutoff? (Methods line 210-211). Other information given beyond genus and species should be defined.

Validity of the findings

1. The identities of the putative compounds presented in tables 1 and 2 are not well supported by the data provided and the process by which these particular compounds were chosen as IDs for these M/Z values is not adequately explained. MS/MS data and analysis of isotopic patterns should be used for putative compound ID, not just M/Z values as done here. The fact that many of the compounds identified are from plants or Bryozoa rather than bacteria indicates that the IDs are probably incorrect. Further, it is unclear why multiple ions are given the same compound ID and same "tolerance" in tables 1 and 2 when they have different M/Z values.
2. The first paragraph of the discussion (lines 262-270) speculates on highly specific mechanisms to explain growth phase -dependent antibiotic observed in this study. No data presented in this paper supports any particular mechanism, so a more general discussion of production and the function of growth phase would be more appropriate.
3. The fact that both strains were identified as Streptomyces, and the significance of each belonging to a different clade should be discussed. (Lines 296-302)

Reviewer 2 ·

Basic reporting

Ahmad et al. reported anti-malarial activity of Streptomyces bacterial extract and attempted to isolate potent metabolites that might be associated with the drug-like activity. Authors approach to investigate anti-malarial properties of bacterial extract is intriguing. However, authors need to improve substantially the textual content, specifically English needed to be improved throughout the manuscript. Additionally, I have two other recommendations:

1) The resolution of figures 4 to 6 is very poor. It is very hard for reviewer to read the contents of the figures.

2) Please expand all the figure legends and explain the contents for readers.

Experimental design

How many independent biological replicates have been performed for generating bacterial growth curve and anti-plasmodial activity IC50 data ? Authors should write the details in the experimental section.

Validity of the findings

1) Line 223-224 Figure 2. Authors stated that anti-malarial potency of SUK 48 at day 14 is more as compared to day 7. However, in figure 2, both time points are close to zero. I suggest authors need re-plot the graph with correct Y-axis, which can demonstrate the observed small differences.

2) Line 223. Authors need to explain in methods or results section how they evaluated anti-plasmodial activity of the bacterial strains. Currently, what authors describe in section “In vitro antimalarial assay” line 99 is incomplete.


3) The asexual blood stages of 3D7 P. falciparum consists of rings, trophozoites and schizont. Authors need to describe at which stage of in vitro culture they incubated the bacterial extracts and for how long (duration of treatment). Does chloroquine treatment were carried out in similar conditions (similar stages and duration of exposure) ?

4) Are there any common metabolites between the extracts of SUK48 and SUK7?

---

## Round 0.2 · Major Revisions

Dear Dr. Ahmad and colleagues:

Thanks for resubmitting your revised manuscript. As you can see from the re-reviews, there are a few more issues that need attention.

I agree with the concerns of the reviewers, and thus feel that their suggestions should be adequately addressed before moving forward. The concerns of reviewer 1, both from this and the prior review, need to be addressed. Also, please improve the clarity of your figures.

I look forward to seeing your revision, and thanks again for submitting your work to PeerJ.

Good luck with your revision,

-joe

Reviewer 1 ·

Basic reporting

1. English language usage needs additional revision before publication
2. Figures are improved, but figure 3 in particular still has lots of illegible data labels and extraneous text
3. In supplementary table 1, these compounds still need to be designated as "potential compounds" as they were not identified unambiguously

Experimental design

1. As noted in my original review, the authors do not actually describe how metabolomics was used to PRIORITIZE a particular extract (section beginning wine 246). They describe how metabolomics was used to analyze extracts, but it's not clear how this analysis led them to particular compounds of interest, or what their criteria were for identifying particular interesting compounds.
2. Throughout, the authors assume that they are observing [M + H]+ ions but some could very well be M + Na, or other adducts.

Validity of the findings

1. As described in my original review, I'm concerned that compound identities proposed from mass spectrometry data in this paper are highly speculative and many could be incorrect in the absence of analyzing MS/MS or isotopic pattern data (or NMR or other analytical methods). The putative nature of these compound IDs is now noted in the discussion, but extensive space is devoted to discussing particular putative compounds, which doesn't seem appropriate given how speculative the IDs are.
2. I remain concerned that the explanation for antibiotic production in a particular growth phase at the beginning of the discussion is not supported by any data in this paper

Reviewer 2 ·

Basic reporting

no comment

Experimental design

no comment

Validity of the findings

no comment

Additional comments

I have read the revised version of the manuscript and the point-by-point rebuttal provided by the authors and I feel that most of the comments and suggestions have been well transferred in the current version of the manuscript. However, the resolution of provided figures are very poor. For example, please see figure 3 . It would be difficult for PeerJ readers to read the content of figure 3.

---

## Round 0.3 · Minor Revisions

Dear Dr. Ahmad and colleagues:

Thanks for revising your manuscript. The reviewers are very satisfied with your revision (as am I). Great! However, there are a few minor things to address. Please address these ASAP so we may move towards acceptance of your work.

Best,

-joe

Reviewer 1 ·

Basic reporting

While improved, there are numerous issues with English language usage throughout.

Experimental design

No comment

Validity of the findings

As mentioned in my previous reviews, the authors' assignment of specific compounds based on their metabolomics data is highly speculative. I don't doubt that these compounds are the closest match between their m/z values and compounds in the databases used (Dictionary of Natural Products and Metlin). However, a similar m/z value is scant evidence for assigning a compound and I think it is highly likely that many, if not most, of their putative compound assignments are incorrect. I think this is especially likely because most of the putative compounds mentioned in the text are not known to be bacterially-produced molecules, but rather are known to come from plants, bacteria, or synthetic sources.

I do appreciate that the authors have revised the language in the text to note that these are putative compounds, not definitive assignments. I also appreciate that the authors have noted that definitive identification will require MS/MS and NMR (lines 317-318).
Mention of compounds in lines 293-300 should be revised to note that these are putative compound assignments.

The authors state that supplementary table 1 has been revised to label the compounds instead as "potential compounds," but I do not see the change on the version of supplementary table 1 provided with this revision. The change should be made.

Finally, as noted in my previous reviews, the amount of space dedicated to discussing literature on putatively assigned compounds seems inappropriate given that these might not be the actual compounds after all. My specific recommendation would be to remove the paragraph from lines 301 through 316 (but keep the final sentence of the paragraph about compound identification by MS/MS and NMR).

Reviewer 2 ·

Basic reporting

No comments

Experimental design

No comments

Validity of the findings

No comments

Additional comments

I have read the revised version of the manuscript and the point-by-point rebuttal provided by the authors. Overall, I recommend the manuscript for publication.

---

## Round 0.4 · accepted · Accept

Dear Dr. Ahmad and colleagues:

Thanks for revising your manuscript based on the concerns raised by the reviewers. I now believe that your manuscript is suitable for publication. Congratulations! I look forward to seeing this work in print, and I anticipate it being an important resource for groups studying Streptomyces metabolomics and drug development. Thanks again for choosing PeerJ to publish such important work.

Best,

-joe